# Cloning, Expression, Purification, and Characterization of β-Galactosidase from *Bifidobacterium longum* and *Bifidobacterium pseudocatenulatum*

**DOI:** 10.3390/molecules27144497

**Published:** 2022-07-14

**Authors:** Mingzhu Du, Shuanghong Yang, Tong Jiang, Tingting Liang, Ying Li, Shuzhen Cai, Qingping Wu, Jumei Zhang, Wei Chen, Xinqiang Xie

**Affiliations:** 1School of Food Science and Technology, Jiangnan University, Wuxi 214122, China; 6190112018@stu.jiangnan.edu.cn (M.D.); 6180112106@stu.jiangnan.edu.cn (S.Y.); 2Guangdong Provincial Key Laboratory of Microbial Safety and Health, State Key Laboratory of Applied Microbiology Southern China, Institute of Microbiology, Guangdong Academy of Sciences, Guangzhou 510070, China; jt0925@stu.scau.edu.cn (T.J.); gdim_liangtt@outlook.com (T.L.); liying@gdim.cn (Y.L.); shuzhenwork@163.com (S.C.); wuqp203@163.com (Q.W.)

**Keywords:** β-galactosidase, *Bifidobacterium*, enzymatic properties

## Abstract

Expression and purification of β-galactosidases derived from *Bifidobacterium* provide a new resource for efficient lactose hydrolysis and lactose intolerance alleviation. Here, we cloned and expressed two β-galactosidases derived from *Bifidobacterium*. The optimal pH for BLGLB1 was 5.5, and the optimal temperature was 45 °C, at which the enzyme activity of BLGLB1 was higher than that of commercial enzyme E (300 ± 3.6 U/mg) under its optimal conditions, reaching 2200 ± 15 U/mg. The optimal pH and temperature for BPGLB1 were 6.0 and 45 °C, respectively, and the enzyme activity (0.58 ± 0.03 U/mg) under optimum conditions was significantly lower than that of BLGLB1. The structures of the two β-galactosidase were similar, with all known key sites conserved. When o-nitrophenyl-β-D-galactoside (oNPG) was used as an enzyme reaction substrate, the maximum reaction velocity (*V*_max_) for BLGLB1 and BPGLB1 was 3700 ± 100 U/mg and 1.1 ± 0.1 U/mg, respectively. The kinetic constant (*K*_m_) of BLGLB1 and BPGLB1 was 1.9 ± 0.1 and 1.3 ± 0.3 mmol/L, respectively. The respective catalytic constant (*k*_cat_) of BLGLB1 and BPGLB1 was 1700 ± 40 s^−1^ and 0.5 ± 0.02 s^−1^, respectively; the respective *k*_cat_/*K*_m_ value of BLGLB1 and BPGLB1 was 870 L/(mmol∙s) and 0.36 L/(mmol∙s), respectively. The *K*_m_, *k*_cat_ and *V*_max_ values of BLGLB1 were superior to those of earlier reported β-galactosidase derived from *Bifidobacterium*. Overall, BLGLB1 has potential application in the food industry.

## 1. Introduction

β-galactosidase (E.C.3.2.1.23) is categorized into five different families (GH1, 2, 35, 42 and 59) in the CAZy database http://www.cazy.org/ (accessed on 25 March 2022) according to their sequence, activity, and structure. β-galactosidase with trans glycosylation activity participates in the transglycosylation reaction to produce prebiotic galactooligosaccharides [1,2]. In addition, β-galactosidase can also be used as a biocatalyst to hydrolyze lactose, which is widely used in dairy processing, food processing, and pharmaceutical fields products [3,4,5,6]. For example, whey is a byproduct of cheese production and contains 4–4.5% lactose. However, since lactose is difficult to digest, most of it is discarded, which in turn seriously pollutes the environment. Lactose is hydrolyzed by β-galactosidase, and when its hydrolysis rate reaches 80%, it can be consumed or used as feed, thereby avoiding wastage of resources [5,7,8]. In addition, for humans, β-galactosidase is an important glycoside hydrolase responsible for hydrolyzing dietary lactose to produce galactose and glucose [4]. Some people can synthesize β-galactosidase normally in their intestines, thereby allowing them to digest and utilize lactose. However, the lack of β-galactosidase could cause lactose intolerance [9,10]. An estimated 75% of the global adult population is intolerant to lactose [11]. In recent decades, researchers have become increasingly interested in the acquisition, identification, and characterization of β-galactosidase due to its important applications in different industries.

Considering safety, production cost, and enzyme yield in practical applications, technicians mainly obtain β-galactosidase from microorganisms, such as bacteria, fungi, and yeast [12,13]. In practical applications, different sources of β-galactosidase should be selected according to the environmental conditions. For example, fungal β-galactosidase is mainly used for acidic whey hydrolysis (pH 2.5–5.4). In contrast, β-galactosidase from yeast shows the highest activity at pH 6.0–7.0, which is more suitable for hydrolyzing milk and sweet whey [14]. Specifically, *Bifidobacterium* is considered a good source of β-galactosidase due to its generally recognized associated safety [15].

With sequencing technology development, we can better understand *Bifidobacterium* at the genetic level, and genetic engineering technology is also widely and efficiently used to obtain β-galactosidase [16,17]. This study performed whole-genome sequencing of *Bifidobacterium* isolates and mined sequences encoding β-galactosidase; β-galactosidase from *Bifidobacterium* was cloned heterologously expressed and purified. Besides, the related enzyme properties were assayed. These results showed that β-galactosidase derived from *Bifidobacterium longum* (*B. longum*) 020402 possesses good enzymatic properties and potential industrial application.

## 2. Results and Discussion

### 2.1. Characterization of BLGLB1 and BPGLB1

Based on the whole genome data of *B*. *pseudocatenulatum* 070108 and *B*. *longum* 020402, we found two genes encoding β-galactosidase; in the sequence information shown in Appendix A, BLGLB1 stands for *Bifidobacterium*
*longum* β-galactosidase 1 and BPGLB1 stands for *Bifidobacterium*
*pseudocatenulatum* β-galactosidase 1. The predicted molecular weights of BLGLB1 and BPGLB1 were 79.82 kDa and 78.14 kDa, respectively, which corresponded to the position information of the bands in the sodium dodecyl sulfate-polyacrylamide gel electrophoresis (SDS-PAGE) (Figure 1a). Using a Tecan infinite M200 Microplate Reader to conduct the Bradford method with bovine serum albumin as the standard, the concentrations of BLGLB1, BPGLB1, and E were 34.8 mg/mL, 17.8 mg/mL, and 21.92 mg/mL, respectively. To determine the optimal concentration and reaction time of the enzyme used in the subsequent enzymatic reaction, a series of reactions were performed with o-nitrophenyl-β-D-galactoside (oNPG) as the substrate at 37 °C/pH = 7.0 with different dilution gradients and reaction times. Considering the uniformity of reaction time, we selected 10 min as the enzyme reaction time for subsequent experiments. The absorbance can fall within the standard curve and the amount of enzyme during the reaction. We diluted BLGLB, BPGLB1, and E to 0.87 μg/mL, 0.89 mg/mL, and 0.04 mg/mL, respectively (Appendix A). Besides, we used these concentrations for subsequent experiments.

The optimal temperature for BLGLB1, BPGLB1, and commercial β-galactosidase E was 45 °C, as shown in Figure 1b. With an increase in temperature from 10 to 45 °C, the activity of β-galactosidase increased; in contrast, increasing the temperature from 50 to 80 °C decreased the activity of β-galactosidase. When the temperature dependence of β-galactosidase enzyme activity was measured in the range 30–50 °C, the relative activity of BLGLB1 was more than 90%, but the activity of BPGLB1 and E was less than 80% and 70%, respectively. These results indicated that commercial enzyme E and BPGLB1 are sensitive to temperature changes, and that BLGLB1 was more stable. Most studies have shown that optimal temperature for microorganism-derived β-galactosidase activity ranges from 25 to 60 °C [18,19,20]. Compared with other *Bifidobacterium* strains, such as *B. bifidum* BGN4 (40 °C) [21], *B. longum* 020402 showed a high optimal temperature for β-galactosidase activity (45 °C).

As shown in Figure 1c, the maximum activity of BLGLB1 was observed at pH 5.5; the optimum pH of BPGLB1 was 6.0, and the optimum pH of commercial β-galactosidase E was 7.0. In general, β-galactosidase BPGLB1 and commercial β-galactosidase E are sensitive to the temperature of the enzyme reaction. The pH of the reaction of β-galactosidase BLGLB1 and BPGLB1 was weakly acidic, whereas the commercial β-galactosidase E is more suitable for a neutral environment. BLGLB1 β-galactosidase shows a good enzymatic activity at pH 5.5–6.5, similar to microbial β-galactosidase (pH 3.0–7.5) [19,22]. Overall, when the substrate is oNPG, the enzyme activity of BLGLB1, BPGLB1, and commercial β-galactosidase E can reach 2200 ± 15 U/mg, 0.58 ± 0.03 U/mg, and 300 ± 3.6 U/mg, respectively, under optimal conditions.

The addition of Zn^2+^, Na^+^, Ca^2+^, Mn^2+^ and Li^+^ apparently led to an increase in the activity of β-galactosidase, as shown in Table 1. However, the increased BLGLB1 activity was lower than BPGLB1 and commercial β-galactosidase E activities. On one hand, the use of Mn^2+^ can increase enzyme activity [23], which is not good for BLGLB1. On the other hand, the stability of the metal ions in BLGLB1 promotes its wide use without considering the influence of Zn^2+^, Na^+^, Ca^2+^, Mn^2+^, and Li^+^. On the contrary, Al^3+^ decreased the activities of BLGLB1, BPGLB1, and commercial β-galactosidase E. It is worth noting that some metal ions increased the activity of BPGLB1 and commercial β-galactosidase E but reduced the activity of BLGLB1, such as Co^+^, Ni^+^, Fe^2+^, and Mg^2+^. Similar to previous research results, Fe^2+^ decreased the activity of β-galactosidase [24]. Hsu also showed that Mn^2+^ leads to decreased β-galactosidase activity in *B. longum* CCRC 15708 [25]. Budriene showed that the activity of β-galactosidase derived from *Penicillium canescens* was not sensitive to Na^+^, K^+^, Cu^2+^, Fe^2+^, Mn^2+^, and Mg^2+^ ions in the growth culture [26]. Vidya reported that the presence of Zn^2+^, Cu^2+^, and Fe^2+^ in the medium strongly inhibited β-galactosidase production [27]. These research results show that β-galactosidase from different sources is affected differently by metal ions. When these β-galactosidases are applied to actual production, their respective characteristics must be carefully considered.

### 2.2. Determination of the Kinetic Parameters of BLGLB1 and BPGLB1

The enzyme kinetic curve is suitable for exploring the enzyme reaction rate and various factors affecting the enzyme reaction rate in enzymes’ biological reaction process. The Michaelis–Menten equation is a velocity equation that studies the relationship between the initial velocity of the enzymatic reaction and the substrate concentration of the enzymatic reaction. In this equation, V_max_ represents the maximum reaction velocity and can also represent the enzyme activity. K_m_ indicates the substrate concentration when the enzyme reaction rate is half the maximum reaction rate [28]. As shown in Figure 2, the fitting curve shows the effects of different concentrations of substrates on the hydrolysis rates of BLGLB1, BPGLB1, and commercial β-galactosidase E, respectively. When the substrate was oNPG, the Vmax of BLGLB1 and BPGLB1 was 3700 ± 100 U/mg and 1.1 ± 0.1 U/mg, respectively; and the K_m_ of BLGLB1 and BPGLB1 was 1.9 ± 0.1 mmol/L and 1.3 ± 0.3 mmol/L, respectively (Figure 2a,b). Therefore, the ability of each BLGLB1 molecule to catalyze the conversion of oNPG to product per unit time is stronger than that of BPGLB1 when the substrate concentrations are similar. The catalytic constant of an enzyme, k_cat_, indicates how many substrate reactions could be catalyzed by each enzyme molecule or enzyme active center per second. The k_cat_/K_m_ was used to measure the catalytic efficiency of the enzyme, also known as the specificity constant. The k_cat_ of BLGLB1 and BPGLB1 was 1700 ± 40 s^−1^ and 0.5 ± 0.1 s^−1^, respectively. Besides, the k_cat_/K_m_ of BPGLB1 was 0.36 L/(mmol∙s), and BLGLB1 showed higher catalytic efficiency, with a k_cat_/K_m_ of 870 L/(mmol∙s) (Table 2). Compared with the β-galactosidase derived from different *Bifidobacterium* strains, BLGLB1 shows advantages in terms of K_m_, k_cat_, and V_max_ (Table 2).

In order to explore the commercial application value of BLGLB1, we determined the relevant kinetic parameters of the commercial β-galactosidase E. When the substrate was oNPG, the V_max_, K_m_ of the commercial β-galactosidase E was 360 ± 20 U/mg and 0.5 ± 0.04 mmol/L, respectively (Figure 2c), which indicated that BLGLB1 hydrolyzed oNPG with higher enzymatic activity compared with commercial β-galactosidase E. In addition, we continued to explore the ability of BLGLB1 to hydrolyze lactose. When the substrate was lactose, the V_max_ and K_m_ of BLGLB1 was 1.5 ± 0.2 U/mg and 2.0 ± 0.16 mmol/L, respectively (Figure 2d), and, consistent with previous findings, β-galactosidases from GH42 family frequently better hydrolyze the synthetic substrate oNPG than lactose [29]. This property is related to the fact that *Bifidobacterium* is well adapted to the degradation of many glycans, especially those in the diet and mucins [30]. Meanwhile, when the substrate was lactose, the V_max_ and K_m_ of the commercial β-galactosidase E was 2.9 ± 0.7 U/mg and 5.34 ± 0.23 mmol/L, respectively (Figure 2e). Compared with BLGLB1, the commercial β-galactosidase E had a better substrate affinity to lactose.

**Table 2 molecules-27-04497-t002:** Enzymatic properties of β-galactosidase from different sources.

Resource	*K*_m_ (mM)	*k* _cat_	*k*_cat_/*K*_m_	*V*_max_ (U∙mg^−1^)	Optimal Conditions	Reference
*B. longum*	0.85 (oNPG)	-	-	70.67 (oNPG)	pH = 7.0/50 °C	[25]
*B. bifidum*	7.0 (oNPG) 24.0 (lactose)	6900 (oNPG) 0.7 (lactose)	985.71 (oNPG) 0.03 (lactose)	-	pH = 6.0/40 °C	[31]
*B*. *adolescentis*	2.5 (oNPG) 3.7 (lactose)	-	-	107 (oNPG) 22 (lactose)	pH = 7.0/37 °C	[32]
*B. longum* subsp. *infantis*	16 ± 2 (lactose)	97 ± 3 (lactose)	6.1 ± 0.5 (lactose)	-	-	[33]
*B. longum* subsp. *infantis*	44 ± 10 (lactose)	8.6 ± 0.1 (lactose)	0.19 ± 0.00 (lactose)	-	-	[33]
*B. longum* subsp. *infantis*	-	-	0.08 ± 0.00 (lactose)	-	-	[33]
*B. animalis* subsp. *lactis*	25.0 (lactose)	-	-	-	pH = 6.5/37 °C	[34]
*Lactobacillus acidophilus*	3.84 (oNPG) 88.98 (lactose)	-	-	-	pH = 6.0/37 °C	[35]
infant feces	20.95 ± 2.76 (oNPG) 140.2 ± 17.7 (lactose)	-	-	5004.50± 318.8 (oNPG) 293.1 ± 14.7 (lactose)	pH = 6.5/50 °C	[36]
*B*. *adolescentis*	60 (pNPG)	-	-	1.129 (lactose)	pH = 6.0/50 °C	[37]
*B. longum*	1.9 ± 0.1 (oNPG)	1700 ± 40 (oNPG)	870 (oNPG)	3700 ± 100 (oNPG)	pH = 5.5/45 °C	This study (BLGLB1)
*B. pseudocatenulatum*	1.3 ± 0.3 (oNPG)	0.5 ± 0.02 (oNPG)	0.36 (oNPG)	1.1 ± 0.1 (oNPG)	pH = 6.0/45 °C	This study (BPGLB1)

The concentrations of enzymes BLGLB1, BPGLB1, and commercial β-galactosidase E were 0.87 μg/mL, 0.89 mg/mL, and 0.044 mg/mL, respectively. The reaction was carried out under their optimal conditions.

### 2.3. Stability of BLGLB1 and BPGLB1

Under certain circumstances, once the non-covalent force that maintains the active protein structure disappears, it will cause partial extension of the molecule to destroy the active site, resulting in molecular inactivation [38]. Considering the practical application, we tested the stability of BLGLB1 and BPGLB1, and used a commercial enzyme as a positive control.

The relative activity of BLGLB1 was weakened after different temperature treatments. When we set the temperature condition to 35 °C, with the increase in incubation time, the relative activity of BLGLB1 gradually increased and reached a peak (75%, 35 degrees, 60 min); after that, the relative activity of BLGLB1 gradually decreased. However, compared to other temperature conditions (50 °C, 55 °C, 60 °C), BLGLB1 still showed higher relative activity at 35, 40, and 45 °C (Figure 3a). BPGLB1 showed obvious protein precipitation at 50 °C and above, which made determination difficult; therefore, only data at 35, 40, and 45 °C were retained. The relative activity of BPGLB1 was 100% or higher after a short treatment (within 30 min) and even after treatment at 35 °C and 40 °C for 10 min, the enzyme activity was higher (more than 2-fold) than that without treatment (Figure 3b). After treatment, the change trend of the relative activity of the commercial β-galactosidase E was similar to that of BLGLB1. However, the commercial β-galactosidase E maintained more than 80% relative activity at 35 °C and 40 °C and its relative activity was approximately 10% higher than that of the untreated enzyme after treatment at 35 °C for 60 min (Figure 3c).

BLGLB1 showed higher pH stability than BPGLB1 and commercial β-galactosidase E at different pH (Figure 3d). BLGLB1 maintained almost 110% or higher relative activity in the pH range of 4.0–9.0. During this range, the relative activity of BLGLB1 was higher than that without treatment, reaching a maximum of 138% at pH 7.0. The relative activity of commercial β-galactosidase E decreased significantly after treatment. However, the relative activity of commercial β-galactosidase E gradually recovered to approximately 50% (pH = 9.0) with the increase of hydroxide ions in the solution. Besides, the residual activity of BPGLB1 was about 80% at pH 4.0–7.0; after that, BPGLB1’s residual activity decreased sharply at pH > 7.0, and the residual enzyme activity was less than 10% at pH = 9.0.

In general, after a short treatment at a low temperature (10–30 min, 35–40 °C), the relative activity of BPGLB1 was improved considerably. After exposure to low temperature for a long duration (35 °C, 60 min), the relative activity of E could be increased by approximately 15%. Although BLGLB1 was not suitable for any treatment, it showed good pH tolerance, especially in the pH range of 4.0–8.0. BPGLB1 and E were intolerant to pH changes and their relative activities were affected.

### 2.4. Evaluation of the Hydrolytic Activity of BLGLB1 and BPGLB1

In order to explore the specificity of BLGLB1 and BPGLB1, we selected five compounds to test their hydrolysis ability. The results are shown in Table 3. When the substrate was pNPG, BLGLB1 (4600 ± 24 U/mg) and BPGLB1 (4.1 ± 0.4 U/mg) showed the highest activities. However, the commercial β-galactosidase E had high hydrolytic activity on oNPG with 300 ± 3.6 U/mg. In addition, BLGLB1 also had high hydrolytic activity on oNPG, and its activity was 2200 ± 15 U/mg. Differently, the ability of BLGLB1 to hydrolyze 4-nitrophenol-α-galactoside (100 ± 10 U/mg) and 2-nitrophenol-β-glucoside (12 ± 3 U/mg) was significantly lower than that of pNPG and oNPG, which showed that BLGLB1 and pNPG have the better binding ability. Furthermore, the hydrolysis capacity of BLGLB1 on lactose was comparable to that of commercial β-galactosidase E, the hydrolysis activity of BPLGB1 on lactose was 1.3 ± 0.1 U/mg, and the hydrolytic activity of E on lactose was 1.6 ± 0.1 U/mg, which means that BLGLB1 has the potential for application. To explore the application of BLGLB1 in natural samples, we chose lactose-containing milk and acidic whey as substrates, and selected commercial β-galactosidase E as the positive control. The results showed that the specific activity of BLGLB1 in the degradation of lactose in milk and acidic whey was 2.3 ± 0.2 U/mg and 4.8 ± 0.8 U/mg, respectively. The specific activity of E to degrade lactose in milk and acidic whey was 3.1 ± 0.5 U/mg and 11 ± 0.3 U/mg, respectively.

Overall, although the ability of BLGLB1 to hydrolyze compounds was better than that of commercial β-galactosidase E, the ability of commercial β-galactosidase E to hydrolyze the actual samples was better than that of BLGLB1, which could be due to the actual samples being mixtures. Various factors, such as additives in commercial milk and the acidic whey’s pH value, could affect the enzyme’s activity. Therefore, to obtain a more active and more stable *Bifidobacterium longum*-derived β-galactosidase, the preparation method of BLGLB1 should continue to be optimized based on the application in the follow-up experiments.

### 2.5. Bioinformatic Analysis

The β-galactosidase genes BLGLB1 and BPGLB1 were sequenced, showing that their primary structures were composed of 719 and 696 amino acids, respectively. Sequence analysis and structure prediction of BLGLB1 and BPGLB1 were performed to analyze the differences and similarities between the two enzymes from the perspective of sequence. The comparison of the sequences of these two enzymes in the NCBI protein Basic Local Alignment Search Tool (Appendix A) showed that the sequences of the enzymes already existed in the database, but without relevant characterization. A comparison of the conserved domains of the two proteins showed that the structure of enzyme BLGLB1 corresponded to β-galactosidase from the GH42 family (sequence 25–398) and the domain where β-galactosidase forms several polymers (sequence 411–622) (Figure 4a). The sequence of BPGLB1 was also divided into two parts: sequence 23–396 corresponded to β-galactosidase of the GH42 family and the other part corresponded to the type 1 glutamine amidotransferase-like domain (sequence 409–619) (Figure 4b).

The primary sequences of BLGLB1 and BPGLB1 were compared with those of other β-galactosidases of the GH42 family, whose sequences belonged to *Bifidobacterium* and were recorded in the GH42 family in the Cazy database. The alignment of such sequences showed a series of conserved amino acid residues (Appendix A), especially P28, D39, A46, G47, N49, W58, D75, D78, T93, P97, W99, P106, G114, G120, R122, P130, N160, E161, N193, A195, W201, P214, F235, D238, T262, D288, Y290, W329, G339, A350, G352, D354, E369, G393, W424, D457, P460, Y468, P474, G494, G495, D507, P518, G530, A601, G610, L619, and G658, which were 100% conserved. Among these conserved amino acid residues, E161 and E321 were the key catalytic sites. These two key catalytic sites were conserved in BLGLB1 and BPGLB1. Moreover, compared with ABI35985.1 (Figure 4c and Appendix A), the β-galactosidase of *Thermus thermophiles* in UniProt [39] showed that R102 and N160 might be the binding sites, Y170 might be the metal-binding site, and N332 might be the active site. However, there was a significant difference in the activities of BLGLB1 and BPGLB1 in the above experiment. Therefore, it is possible that other sites in the catalytic domain affect the properties of the enzyme. Among the compared sequences, the similarity between ABE95118.1 and BLGLB1 was as high as 86.90% and the similarity between AAR24113.1 and BPGLB1 was as high as 82.76%. In the sequence alignment of residues 630–640, the C-terminus of the BLGLB1 protein had 20 amino acid residues more than BPGLB1. The results of SWISS-MODEL prediction show that both BLGLB1 and BPGLB1 are trimeric structures, which have 49.42% and 48.39% homology compared to the template protein 4uzs.1.A, respectively (Figure 4d,e).

## 3. Materials and Methods

### 3.1. Materials

Shenggong Bioengineering (Shanghai, China) supplied o-nitrophenyl-β-D-galactoside (oNPG), 4-nitrophenol-β-galactoside (pNPG), 4-nitrophenol-α-galactoside, 2-nitrophenol-β-glucoside, β-galactosidase, isopropyl-beta-D-thiogalactopyranoside (IPTG), lysozyme, protein marker, and deoxyribonucleic acid (DNA) marker. The commercial β-galactosidase (Diamond, Shanghai, China) came from *Escherichia coli*; its molecular weight was 540 kDa. According to the definition of enzyme activity, one unit hydrolyzes one micromole of oNPG per minute at 25 °C, under pH 7.5. Its activity was ≥ 50 units per mg dry weight. High-fidelity enzymes (NEB, Ipswich, MA, USA) and endonucleases were supplied by Zhenzhi Biology (Guangzhou, China). The OMEGA purification kit, gel recovery kit, and plasmid extraction kit were purchased from HuanKai Microbial (Guangzhou, China). All other reagents were of analytical or superior grade and were purchased from Huankai Microbial Technology (Guangzhou, China).

*E. coli* BL21 was used as the expression host and was purchased from Shenggong Bioengineering (Shanghai, China). The pET28a plasmid was used as an expression vector to construct a plasmid, which was preserved by the Microbiological Safety and Health Team of the Institute of Microbiology, Guangdong Academy of Sciences. *E. coli* DH5α purchased from Shenggong Bioengineering (Shanghai, China) was used for plasmid construction.

### 3.2. Bacterial Strains

*B*. *pseudocatenulatum* 070108 was isolated from stool samples collected from three healthy fecal microbiota transplantation donors at the First Affiliated Hospital of Guangdong Pharmaceutical University [40]. *B*. *longum* 020402 was isolated from human fecal samples collected in Jiaoling, Guangdong, China. The specific steps of *Bifidobacterium* strains isolation and identification are as follows: we isolated *Bifidobacterium* strains from different fecal sources using a selective isolation medium [40]. First, the fecal samples were diluted with 0.9% SPSS to obtain serial bacterial suspension (10-fold) from 10^−1^ to 10^−6^ dilution; 100 μL of this was cultured on selective agar plates until dry and cultured with 3 parallels under anaerobic conditions at 37 °C for 48 h. According to the different colonial morphologies, single colonies were selected randomly and streaked on selective agar plates twice. After that, single colonies were inoculated in TPY broth for cultivation until obvious turbidity was observed, collected, and identified by Sanger sequencing.

### 3.3. Whole Genome Sequencing and β-Galactosidase Gene Mining

Bacterial DNA was extracted using the HiPure Microbial DNA Kit (Magen Biotech, Guangzhou, China) according to the manufacturer’s instructions. Whole genome data were obtained using the Illumina NextSeq instrument owned by the Institute of Microbiology, Guangdong Academy of Sciences. Based on whole genome data of *B*. *pseudocatenulatum* 070108 and *B*. *longum* 020402, we used R language to annotate whole genome data and mined gene sequences encoding *β-galactosidase.* ProtPara https://web.expasy.org/protparam/ (accessed on 25 March 2022) was used for the prediction of physical and chemical properties.

### 3.4. Cloning and Expression of BLGLB1 and BPGLB1

Based on whole genome data obtained from previous experiments, we designed two pairs of degenerate primers to clone the *β-galactosidase* gene (BLGLB1-derived *B. longum* 020402: forward primer 5′-AAAAAACATATGACTACTCGTAGAACGTTCAGG-3′ and reverse primer 5′-GTGGTGCTCGAGTTAGCAGGACGTTTTAGCG-3′; BPGLB1-derived *B*. *pseudocatenulatum* 070108: forward primer 5′- AAAAAAGCTAGCATGCATCGCACGTTCAAATGG-3′ and reverse primer 5′- GTGGTGCTCGAGTTATGCGCGCTTTACGACGAG-3′). We used the genomic DNA (gDNA) of *B*. *pseudocatenulatum* 070108 and *B*. *longum* 020402 as a template for polymerase chain reaction (PCR) amplification, and the PCR product was cloned into the expression plasmid pET28a. The resulting vectors (pET28a/β-galEa) were transformed into *E*. *coli* DH5α and the positive clones were transformed into *E*. *coli* BL21.

*E. coli* BL21 harboring pET28a/β-galEa was inoculated in 500 mL of Super Broth containing 50 mg/L kanamycin and grown for 6 h. The culture was cooled to 18 °C for 30 min, and then we added 400 μL IPTG (1 mol/L) to the culture. The culture was incubated for 48 h, and shaking was required for all incubations. After that, we collected bacterial cells via centrifugation at a low temperature and high speed (9000× *g*, 4 °C, 20 min). Then, the cells were resuspended in a 50 mL centrifuge tube with 35 mL of lysis buffer; lysozyme (0.2 g) was added for lysis and stored at −80 °C until use.

### 3.5. Purification of BLGLB1 and BPGLB1

According to Xie’s methods [41], the protein purification steps were as follows: The bacterial cells were taken out of the refrigerator at −80 °C and thawed at 25 °C; this was followed by intermittent sonication (600 W, 40–50 min) until the bacterial suspension was no longer turbid. The mixed liquid was transferred to a high-speed centrifuge tube for low-temperature, high-speed centrifugation (4 °C, 24,000× *g*, 30 min) to separate the supernatant and cells. The supernatant was carefully aspirated and filtered using 0.8-μm filters. A 5-mL Ni^2+^ affinity column (HisTrap™ FF; GH Healthcare, Chicago, IL, USA) was equilibrated with approximately 50 mL of lysis buffer (1 M NaCl, 50 mM NaH_2_PO_4_, 70 mM Na_2_HPO_4_, and 50 mM imidazole, pH 7.6–7.8). The supernatant was loaded onto a pre-charged column at a rate of 3 mL/min. The column was washed with 25 mL of lysis buffer and then with 25 mL of washing buffer (50 mM sodium phosphate, 1 M NaCl, 60 mM imidazole, pH 7.6) at a rate of 5 mL/min to remove the contaminated proteins. Proteins were eluted from the column using an elution buffer (150 mmol/L NaCl, 150 mmol/L NaH_2_PO_4_, 50 mmol/L Na_2_HPO_4_, and 200 mmol/L imidazole, pH 7.2). The purified protein was dialyzed against 50 mL of exchange buffer (100 mmol/L NaCl, 100 mmol/L NaH_2_PO_4_, and 50 mmol/L NaH_2_PO_4_, pH 7.2) three times. Finally, the buffer was exchanged with the stored buffer (50 mM sodium phosphate, 10% glycerol, 100 mM NaCl, pH 7.2), concentrated, and stored at −80 °C until use. Sodium dodecyl sulfate-polyacrylamide gel electrophoresis (SDS-PAGE) was used to verify the purity of the target protein.

### 3.6. Quantification of β-Galactosidase Activity

The activity of β-galactosidases (BLGLB1 and BPGLB1) was measured by using oNPG (2 mM) dissolved in a phosphate buffer solution (pH 7.0, 10 mM) as the substrate [6]. One international activity unit was defined as the amount of β-galactosidase that hydrolyzed 1 μmoL oNPG per minute. The oNPG hydrolysis rate was expressed as the amount of *o*-nitrophenol (oNP) produced, which was measured using a spectrophotometer at 420 nm. All measurements were performed in triplicates. The protein concentration of the enzyme was quantified according to the manufacturer’s instructions. The commercial β- galactosidase (Diamond, Shanghai, China) named “E” was used as a positive control in all experiments. The average and standard deviation of each determination were recorded. For correlation analysis, a significance level (α) of 0.05 and an analysis of variance were used to check the results.

### 3.7. Effect of Temperature and pH on BLGLB1 and BPGLB1 Activity

To determine the optimal reaction temperature of β-galactosidase, we designed the following protocols based on previous research [42], we prepared the substrate (2 mM oNPG, pH = 7) at different temperatures (20, 25, 30, 35, 40, 45, 50, 60, 70, and 80 °C), adding β-galactosidase to react for 10 min, then adding sodium carbonate to stop the reaction, and then measuring the absorbance at 420 nm to calculate the enzyme activity. The optimum reaction temperature is the maximum enzyme activity. Based on the above operation, the optimal temperature reaction conditions were obtained. Then, 20 mM oNPG was diluted to 2 mM with different pH (3, 4, 5, 5.5, 6, 6.5, 7, 8, and 9), incubated at the optimal temperature, and then β-galactosidase was added to react for 10 min. Then, sodium carbonate was added to terminate the reaction. The absorbance was measured at 420 nm to obtain the optimal reaction pH for β-galactosidase.

### 3.8. Effect of Metallic Cations on BLGLB1 and BPGLB1 Activity

To explore the effect of metal ions on the β-galactosidase, we referred to previous studies [43], and used 2 mM oNPG as the substrate, different metal ions (Zn^2+^, Co^2+^, Ni^2+^, Al^3+^, Li^2+^, Fe^2+^, Mn^2+^, Mg^2+^, Na^+^, and Ca^2+^) were added to a final concentration of 5 mmol/L, and then we measured the enzyme activity under optimal reaction conditions (temperature and pH) determined in Section 3.7.

### 3.9. Determination of the Kinetic Parameters of BLGLB1 and BPGLB1

According to Liao’s methods [28], we used oNPG with a concentration of 0.1–10.0 mol/mL as the substrate, and the experiment was repeated three times. The activity of β-galactosidase was measured under optimal conditions. The observed rate and concentration of the substrate were processed using OriginPro^®^ 8.5 and fitted with nonlinear least square regression. Finally, the fitted Hill function (*n* = 1) was used to calculate the e kinetic constant (*K*_m_) and maximum reaction velocity (*V*_max_).

### 3.10. Stability

According to previous research [42], the enzyme was mixed with buffers of different pH (3.0–9.0), and the mixture was placed in a water bath at 37 °C for 1 h to determine pH stability. By exposing the enzyme to different temperatures (10–80 °C) for different durations (10, 20, 30, 60, and 120 min), temperature stability was determined. All enzyme activities were measured under optimal conditions. The enzyme activity of sample without any treatment was defined as 100% and the remaining enzyme activity measured after treatment under different conditions was converted into relative activity.

### 3.11. Evaluation of the Hydrolytic Activity of BLGLB1 and BPGLB1

Under optimal reaction conditions, we used several substrates (oNPG, pNPG, 4-nitrophenol-α-galactoside and 2-nitrophenol-β-glucoside, 40 mM; lactose, 200 mM; milk; acidic whey) to evaluate the hydrolytic activity of β-galactosidase with different sources. The milk used was ordinary sterilized milk sold in supermarkets. Acidic whey was the supernatant obtained by centrifugation after *Lactobacillus plantarum* fermentation, and its pH was 3.67.

The ability of β-galactosidases to hydrolyze oNPG, pNPG, 4-nitrophenol-α-galactoside and 2-nitrophenol-β-glucoside was assessed using the method described in Section 3.7.

Besides, β-galactosidases hydrolyze the substrate (lactose and lactose in milk) to produce glucose, so we used the 3,5-dinitrosalicylic acid (DNS) assay to assess lactose hydrolysis capacity [44]. Its reaction principle is that DNS undergoes a redox reaction by reducing sugar under alkaline conditions to generate 3-amino-5-nitrosalicylic acid, which appears brown-red under boiling conditions. The color depth is proportional to the reducing sugar content within a specific concentration range. In this step, we first established a standard curve for glucose content, and the reaction system was visible, as shown in the Appendix A. Different enzyme concentrations (40 μL) were reacted with the substrate (160 μL). After 10 min of reaction under optimal conditions, the mixture was placed in a 100 °C water bath, 600 μL of DNS solution was added and boiled for 10 min, and the mixture was allowed to cool and diluted 5 times. Then the absorbance value was measured at 540 nm using a Tecan infinite M200 Microplate Reader (Thermo Fisher Scientific, Waltham, MA, USA) to determine the glucose content generated by the reaction and further determine the hydrolysis efficiency of the enzyme.

### 3.12. Bioinformatic Analysis

The conserved domains were analyzed using the conserved domains in the National Center for Biotechnology Information (NCBI) (https://www.ncbi.nlm.nih.gov/Structure/cdd/wrpsb.cgi, accessed on 25 March 2022). Sequence alignment and analysis were performed using the online software CLUSTALW (https://www.genome.jp/tools-bin/clustalw, accessed on 25 March 2022), and ESPript 3 (https://espript.ibcp.fr/ESPript/cgi-bin/ESPript.cgi, accessed on 25 March 2022). All sequences involved in alignment were obtained from the carbohydrate active enzyme database and UniProt.

## 4. Conclusions

In this study, we cloned and expressed two β-galactosidases from *Bifidobacterium* belonging to the GH42 family. BLGLB1 was composed of 719 amino acids, with an optimal pH of 5.5 and an optimal temperature of 45 °C. Under optimal conditions (substrate is oNPG), the enzyme activity could reach 2200 ± 15 U/mg, significantly higher than that of the commercial enzyme E (300 ± 3.6 U/mg). BPGLB1 composed 697 amino acids; the optimal pH was 6.0, and the optimal temperature was 45 °C. Its enzymatic activity under optimal conditions (substrate is oNPG) reached 0.58 ± 0.03 U/mg, significantly lower than that of BLGLB1. At the same time, a comparison with the enzyme kinetic parameters of known β-galactosidase showed that BLGLB1 has a good application potential.

Based on the whole genome sequencing data of *Bifidobacterium* strains, the bioinformatic analysis of the two β-galactosidases showed that their structures were similar, and all known key sites were conserved. The biggest difference was that BLGLB1 contained an additional 20-amino acid sequence at the C-terminus. However, it is worth elucidating whether this sequence affected enzyme activity. Besides, we successfully obtained two β-galactosidases of the GH42 family by constructing recombinant plasmids for heterologous expression and purification. However, we need to elucidate the active sites for the differences in the activities of the two β-galactosidase enzymes belonging to the same family of *Bifidobacterium*. In addition to β-galactosidase, many types of GHs in *Bifidobacterium* have research significance and application value. For example, fucose hydrolase can hydrolyze human milk oligosaccharides, and oligomeric-1,6-glucosidase can improve the utilization rate and production efficiency of starch raw materials. In addition, we need to explore specific ways to apply the experimental results obtained in the laboratory to real life in the future.

## Figures and Tables

**Figure 1 molecules-27-04497-f001:**
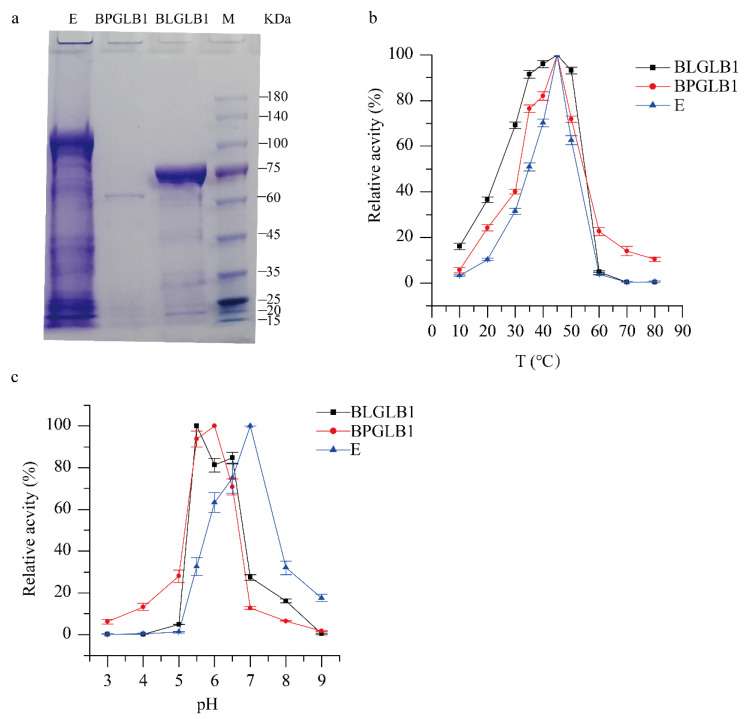
Characterization of BLGLB1 and BPGLB1. (**a**) The SDS-PAGE of BLGLB1, BPGLB1, and E; (**b**) The effect of temperature on enzyme activity; (**c**) The effect of the pH on enzyme activity. The concentrations of BLGLB1, BPGLB1, and commercial β-galactosidase E were 0.87 μg/mL, 0.89 mg/mL, and 0.04 mg/mL, respectively.

**Figure 2 molecules-27-04497-f002:**
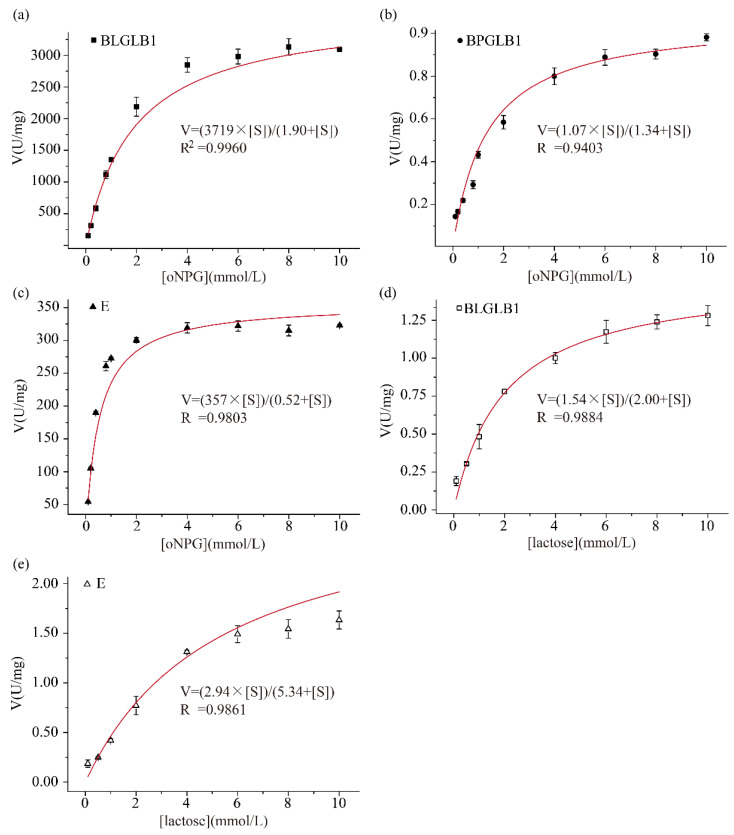
Kinetic parameters. Different concentrations of oNPG were used as the substrate: (**a**) The kinetic parameters of BLGLB1; (**b**) BPGLB1; (**c**) E; different concentrations of lactose were used as the substrate: (**d**) BLGLB1; (**e**) E. The determination was carried out under the respective optimal reaction conditions. The concentration of BLGLB1, BPGLB1, and commercial β-galactosidase E was 0.87 μg/mL, 0.89 mg/mL and 0.04 mg/mL, respectively.

**Figure 3 molecules-27-04497-f003:**
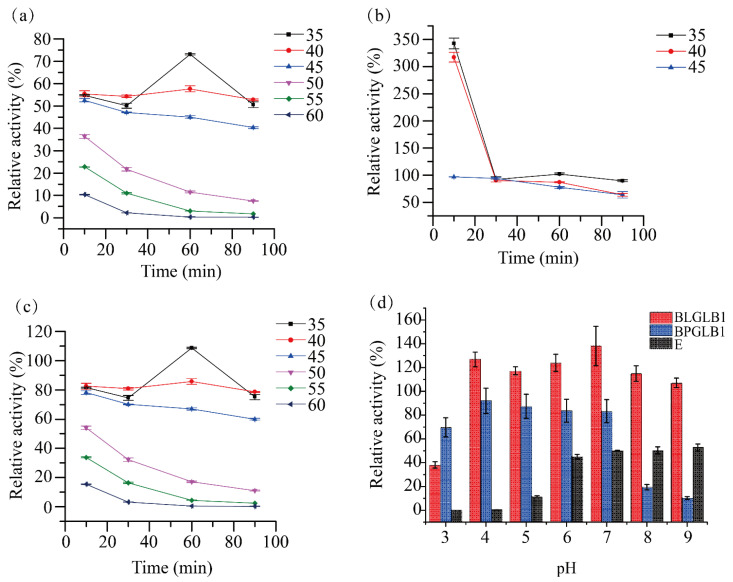
The temperature and pH stability. (**a**) The temperature stability of BLGLB1; (**b**) BPGLB1; (**c**) E; (**d**) the acid-base stability of BLGLB1, BPGLB1, and E. The numbers in the legends in (**a**–**c**) are the reaction temperature (°C), all reactions are carried out under the optimal reaction conditions of each of the three enzymes. The concentrations of BLGLB1, BPGLB1, and commercial β-galactosidase E were 0.87 μg/mL, 0.89 mg/mL, and 0.04 mg/mL respectively. The activity of untreated BLGLB1, BPGLB1 and E was 2200 ± 15 U/mg, 0.58 ± 0.03 U/mg, and 300 ± 3.6 U/mg, respectively.

**Figure 4 molecules-27-04497-f004:**
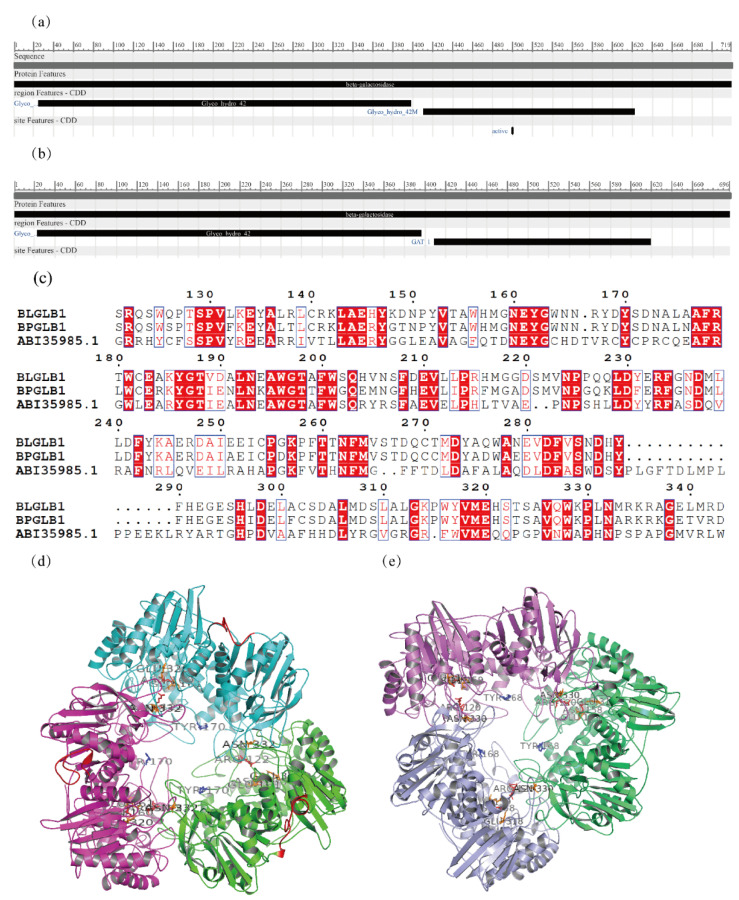
Bioinformatic analysis of BLGLB1 and BPGLB1. (**a**) The key amino acid residue positions of sequence alignment of BLGLB1 with the β-galactosidase belonging to the GH42 family from *Bifidobacterium* in the Cazy database; (**b**) BPGLB1; (**c**) The key amino acid residue positions of sequence alignment of BLGLB1 and BPGLB1 with the β-galactosidase from *Thermus thermophiles*; (**d**) Homology analysis and model prediction of BLGLB1; (**e**) Homology analysis and model prediction of BPGLB1. Both BLGLB1 and BPGLB1 are trimeric structures, and different colors are used to distinguish monomers.

**Table 1 molecules-27-04497-t001:** The effect of metal ions on the activities of BLGLB1, BPGLB1, and E.

Metallic Cations (5 mM)	BLGLB1 (%)	BPGLB1 (%)	E (%)
Untreated	100	100	100
Zn^2+^	103 ± 2.6	122 ± 2.5	147 ± 1.4
Co^2+^	87 ± 4.2	186 ± 5.1	141 ± 1.0
Al^3+^	44 ± 0.1	96 ±1.7	60 ± 3.9
Ni^2+^	83 ± 3.6	212 ± 1.1	156 ± 3.7
Fe^2+^	94 ± 0.5	104 ± 1.8	132 ± 5.6
Mg^2+^	96 ± 2.5	154 ± 1.9	162 ± 6.2
Na^+^	104 ± 0.9	167 ± 2.0	121 ± 3.2
Ca^2+^	104 ± 0.3	111 ± 4.6	104 ± 9.7
Mn^2+^	101 ± 0.6	101 ± 1.1	161 ± 2.0
Li^+^	109 ± 1.1	129 ± 13.2	115 ± 6.3

The concentrations of BLGLB1, BPGLB1 and commercial β-galactosidase E are 0.87 μg/mL, 0.89 mg/mL, and 0.04 mg/mL respectively. The reactions are carried out under the respective optimal conditions.

**Table 3 molecules-27-04497-t003:** Determination of specificity of the enzyme for the three β-galactosidases.

Enzyme	BLGLB1 (U/mg)	BPGLB1 (U/mg)	E (U/mg)
pNPG	4600 ± 24	4.1 ± 0.4	54 ± 3
oNPG	2200 ± 15	0.58 ± 0.03	300 ± 3.6
4-nitrophenol-α-galactoside	100 ± 10	0.4 ± 0.03	0.5 ± 0.4
2-nitrophenol-β-glucoside	12 ± 3	0.07 ± 0.01	0
Lactose	1.3 ± 0.1	-	1.6 ± 0.1
Milk	2.3 ± 0.2	-	3.1 ± 0.5
Acidic whey	4.8 ± 0.8	-	11 ± 0.3

The concentrations of enzymes BLGLB1, BPGLB1, and commercial β-galactosidase E are 0.87 μg/mL, 0.89 mg/mL, and 0.04 mg/mL, respectively. The reaction is carried out under their optimal conditions.

## Data Availability

Data will be available upon reasonable request from the corresponding author.

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
