# Peer review of "Cloning, Expression, Purification, and Characterization of β-Galactosidase from Bifidobacterium longum and Bifidobacterium pseudocatenulatum"

_molecules, 2022, doi:10.3390/molecules27144497_

Round 1
Reviewer 1 Report
The manuscript was improved in relation to the description of the methodology, the main point criticized and as previously mentione.
Author Response
Thank you for your suggestions for this manuscript.
Reviewer 2 Report
The manuscript presents the clonation and expression of two β-galactosidases isolated from Bifidobacterium strains, which can be efficiently applied in diminishing lactose intolerance and hydrolisis.
The article is well prepared, but there are some aspects that should be improved:
line 83 - oNPG - please define the abbreviation in the text also, not only in the abstract.
line 120, 121, and 123 - please cite references according to journal guidelines - revise the whole manuscript
in the results and discussion section; there are some subsections that are not sufficiently described and discussed - i.e., 2.2. Determination of the kinetic parameters of BLGLB1 and BPGLB1; 2.3. Stability of BLGLB1 and BPGLB1; 2.4. Evaluation of the hydrolytic activity of BLGLB1 and BPGLB1.
Also, in the material and method section, each experiment and protocol should be based on similar research and specifications. Please revise.
The grammar and flowability of the article should be revised by a native English speaker.
Otherwise, the manuscript is quite well improved, and after some major revisions, it can be considered for publication.
Author Response
We are very appreciated that the reviewer read our manuscript carefully and provide many important suggestions for improving our work. Based on these questions, we have revised our manuscript substantially as following below.
The manuscript presents the clonation and expression of two β-galactosidases isolated from Bifidobacterium strains, which can be efficiently applied in diminishing lactose intolerance and hydrolisis.
The article is well prepared, but there are some aspects that should be improved. Otherwise, the manuscript is quite well improved, and after some major revisions, it can be considered for publication.
1、line 83 - oNPG - please define the abbreviation in the text also, not only in the abstract.
We have modified this mistake.(line 83)
2、line 120, 121, and 123 - please cite references according to journal guidelines - revise the whole manuscript
We have modified this mistake and revised the whole manuscript. (Lines 121、122、124)
3、in the results and discussion section; there are some subsections that are not sufficiently described and discussed - i.e., 2.2. Determination of the kinetic parameters of BLGLB1 and BPGLB1; 2.3. Stability of BLGLB1 and BPGLB1; 2.4. Evaluation of the hydrolytic activity of BLGLB1 and BPGLB1.
We have refined the results description and discussion, and the revised manuscript are as follows (Lines 137-171、181-210、224-249):
2.2. Determination of the kinetic parameters of BLGLB1 and BPGLB1
The enzyme kinetic curve is suitable for exploring the enzyme reaction rate and various factors affecting the enzyme reaction rate in enzymes' biological reaction process. The Michaelis-Menten equation is a velocity equation that studies the relationship between the initial velocity of the enzymatic reaction and the substrate concentration of the enzymatic reaction. In this equation, Vmax represents the maximum reaction velocity and can also represent the enzyme activity. Km indicates the substrate concentration when the enzyme reaction rate is half the maximum reaction rate [1]. As shown in Figure 2, the fitting curve shows the effects of different concentrations of substrates on the hydrolysis rates of BLGLB1, BPGLB1, and commercial β-galactosidase E, respectively. When the substrate was oNPG, the Vmax of BLGLB1 and BPGLB1 was 3700 ± 100 U/mg and 1.1 ± 0.1 U/mg, respectively; and the Km of BLGLB1 and BPGLB1 was 1.9 ± 0.1 mmol/L and 1.3 ± 0.3 mmol/L, respectively (Figure 2a,b). Therefore, the ability of each BLGLB1 molecule to catalyze the conversion of oNPG to product per unit time is stronger than that of BPGLB1 when the substrate concentrations are similar. The catalytic constant of an enzyme, kcat, indicates how many substrate reactions could be catalyzed by each enzyme molecule or enzyme active center per second. The kcat/Km was used to measure the catalytic efficiency of the enzyme, also known as the specificity constant. The kcat of BLGLB1 and BPGLB1 was 1700 ± 40 s-1 and 0.5 ± 0.1 s-1, respectively. Besides, the kcat/Km of BPGLB1 was 0.36 L/(mmol∙s), and BLGLB1 showed higher catalytic efficiency, with a kcat/Km of 870 L/(mmol∙s) (table 2). Compared with the β-galactosidase derived from different Bifidobacterium strains, BLGLB1 shows advantages in terms of Km, kcat, and Vmax (Table 2).
In order to explore the commercial application value of BLGLB1, we determined the relevant kinetic parameters of the commercial β-galactosidase E. When the substrate was oNPG, the Vmax, Km of the commercial β-galactosidase E was 360 ± 20 U/mg and 0.5 ± 0.04 mmol/L, respectively (figure 2c), which indicated that BLGLB1 hydrolyzed oNPG with higher enzymatic activity compared with commercial β-galactosidase E. In addition, we continued to explore the ability of BLGLB1 to hydrolyze lactose. When the substrate was lactose, the Vmax and Km of BLGLB1 was 1.5 ± 0.2 U/mg and 2.0 ± 0.16 mmol/L, respectively (Figure 2d), consistent with previous findings, β-galactosidases from GH42 family frequently hydrolyze better the synthetic substrate oNPG than lactose [2]. This property is related to the fact that Bifidobacterium is well adapted to the degradation of many glycans, especially those in the diet and mucins [3]. Meanwhile, when the substrate was lactose, the Vmax and Km of the commercial β-galactosidase E was 2.9 ± 0.7 U/mg and 5.34 ± 0.23 mmol/L, respectively (Figure 2e). Compared with BLGLB1, the commercial β-galactosidase E had a better substrate affinity to lactose.
2.3. Stability of BLGLB1 and BPGLB1
Under certain circumstances, once the non-covalent force that maintains the active protein structure disappears, it will cause partial extension of the molecule to destroy the active site, resulting in molecular inactivation [4]. Considering the practical application, we tested the stability of BLGLB1 and BPGLB1, and used a commercial enzyme as a positive control.
The relative activity of BLGLB1 was weakened after different temperature treatments. When we set the temperature condition to 35 °C, with the increase in incubation time, the relative activity of BLGLB1 gradually increased and reached a peak (75%, 35 degrees, 60 min); after that, the relative activity of BLGLB1 gradually decreased. However, compared to other temperature conditions (50 °C, 55 °C, 60 °C), BLGLB1 showed still higher relative activity at 35, 40 and 45 °C (Figure 3a). BPGLB1 showed obvious protein precipitation at 50 °C and above, which made determination difficult; therefore, only data at 35, 40 and 45 °C were retained. The relative activity of BPGLB1 was 100% or higher after a short treatment (within 30 min) and even after treatment at 35 °C and 40 °C for 10 min, the enzyme activity was higher (more than 2-fold) than that without treatment (Figure 3b). After treatment, the change trend of the relative activity of the commercial β-galactosidase E was similar to that of BLGLB1. However, the commercial β-galactosidase E maintained more than 80% relative activity at 35 °C and 40 °C and its relative activity was approximately 10% higher than that of the untreated enzyme after treatment at 35 °C for 60 min (Figure 3c).
BLGLB1 showed higher pH stability than BPGLB1 and commercial β-galactosidase E at different pHs (Figure 3d). BLGLB1 maintained almost 110% or higher relative activity in the pH range of 4.0-9.0. During this range, the relative activity of BLGLB1was higher than that without treatment, reaching a maximum of 138% at pH 7.0. The relative activity of commercial β-galactosidase E decreased significantly after treatment. However, the relative activity of commercial β-galactosidase E gradually recovered to approximately 50% (pH = 9.0) with the increase of hydroxide ions in the solution. Besides, the residual activity of BPGLB1 was about 80% at pH 4.0-7.0; after that, BPGLB1’s residual activity decreased sharply at pH>7.0 , and the residual enzyme activity was less than 10% at pH=9.0.
In general, after a short treatment at a low temperature (10-30 min, 35-40 °C), the relative activity of BPGLB1 was improved considerably. After exposure to low temperature for a long duration (35 °C, 60 min), the relative activity of E could be increased by approximately 15%. Although BLGLB1 was not suitable for any treatment, it showed good pH tolerance, especially in the pH range of 4.0-8.0. BPGLB1 and E were intolerant to pH changes and their relative activities were affected.
2.4. Evaluation of the hydrolytic activity of BLGLB1 and BPGLB1
In order to explore the specificity of BLGLB1 and BPGLB1, we selected five compounds to test their hydrolysis ability. The results are shown in Table 3. When the substrate was pNPG, BLGLB1 (4600 ± 24 U/mg), BPGLB1 (4.1 ± 0.4 U/mg) all showed the highest activities. However, the commercial β-galactosidase E had high hydrolytic activity on oNPG with 300 ± 3.6 U/mg. In addition, BLGLB1 also had high hydrolytic activity on oNPG, and its activity was 2200 ± 15 U/mg. Differently, the ability of BLGLB1 to hydrolyze 4-nitrophenol-α-galactoside (100 ± 10 U/mg) and 2-nitrophenol-β-glucoside (12 ± 3 U/mg) was significantly lower than that of pNPG and oNPG, which showed that BLGLB1 and pNPG have the better binding ability. Furthermore, the hydrolysis capacity of BLGLB1 on lactose was comparable to that of commercial β-galactosidase E, the hydrolysis activity of BPLGB1 on lactose was 1.3 ± 0.1 U/mg, and the hydrolytic activity of E on lactose was 1.6 ± 0.1 U/mg, which means that BLGLB1 has the potential for application. To explore the application of BLGLB1 in natural samples, we chose lactose-containing milk and acidic whey as substrates, selected commercial β-galactosidase E as positive control. The results showed that the specific activity of BLGLB1 in the degradation of lactose in milk and acidic whey was 2.3 ± 0.2 U/mg and 4.8 ± 0.8 U/mg, respectively. The specific activity of E to degrade lactose in milk and acidic whey was 3.1 ± 0.5 U/mg and 11 ± 0.3 U/mg, respectively.
Overall, although the ability of BLGLB1 to hydrolyze compounds was better than that of commercial β-galactosidase E, the ability of commercial β-galactosidase E to hydrolyze actual samples was better than that of BLGLB1, which could be due to the actual samples being mixtures. Various factors, such as additives in commercial milk, and the acidic whey's pH value, could affect the enzyme's activity. Therefore, to obtain a more active and more stable Bifidobacterium longum-derived β-galactosidase, the preparation method of BLGLB1 should continue to be optimized based on the application in the follow-up experiments,
4、Also, in the material and method section, each experiment and protocol should be based on similar research and specifications. Please revise.
We have added the information of the method. (Line 378-380、391、403、408、415、434)
5、The grammar and flowability of the article should be revised by a native English speaker.
We have revised the whole manuscript.
- Liao, F.; Tian, K.C.; Yang, X.; Zhou, Q.X.; Zeng, Z.C.; Zuo, Y.P. Kinetic substrate quantification by fitting the enzyme reaction curve to the integrated Michaelis-Menten equation. ANALYTICAL AND BIOANALYTICAL CHEMISTRY 2003, 375, 756-762, doi:10.1007/s00216-003-1829-x.
- Hildebrandt, P.; Wanarska, M.; Kur, J. A new cold-adapted beta-D-galactosidase from the Antarctic Arthrobacter sp. 32c - gene cloning, overexpression, purification and properties. BMC Microbiol 2009, 9, 151, doi:10.1186/1471-2180-9-151.
- Turroni, F.; Bottacini, F.; Foroni, E.; Mulder, I.; Kim, J.H.; Zomer, A.; Sánchez, B.; Bidossi, A.; Ferrarini, A.; Giubellini, V.; et al. Genome analysis of Bifidobacterium bifidum PRL2010 reveals metabolic pathways for host-derived glycan foraging. Proc Natl Acad Sci U S A 2010, 107, 19514-19519, doi:10.1073/pnas.1011100107.
- Fernandez-Lopez, L.; Pedrero, S.G.; Lopez-Carrobles, N.; Gorines, B.C.; Virgen-Ortiz, J.J.; Fernandez-Lafuente, R. Effect of protein load on stability of immobilized enzymes. ENZYME AND MICROBIAL TECHNOLOGY 2017, 98, 18-25, doi:10.1016/j.enzmictec.2016.12.002.

Reviewer 3 Report
The authors revised their manuscript based on my comments. To me, this paper is acceptable.
Author Response

(The authors gave the same response as above.)

Round 2
Reviewer 2 Report
The article has been corrected according to requirements. As an additional correction, please define more keywords, for the manuscript, as this is also an important element.
This manuscript is a resubmission of an earlier submission. The following is a list of the peer review reports and author responses from that submission.
Round 1
Reviewer 1 Report
Title: Cloning, Expression, Purification, and Characterization of β- Galactosidase from Bifidobacterium longum and Bifidobacterium pseudocatenulatum
The article deals with the characterization of β- Galactosidases from two Bifidobacterium species. Although the paper sounds within the scope of the journal Molecules, there are a lot of problems concerning the scientific and technical aspects of the manuscript. Along with these aspects, in some topics, the English expression of the manuscript is awful and needs to be revised.
Due to these problems, I do not recommend the publication of this manuscript.
Below I will give some examples to support my recommendation. However, it must be realized that I could have made many more comments than I did.
(1) First, it is necessary to clarify better the subject of the manuscript. The introduction of the manuscript is too much of a collection of loose facts and too little about state-of-the-art. In my opinion, this is not acceptable. I suggest that the authors really should take the care to present the "introduction" properly, with a proper background and argument to improve the manuscript, especially thinking about the audience of Molecules.
(2) The materials & methods section does not describe in detail how the experiments were done. Indeed, I would say that, in some cases, it does not describe the procedure at all. If the materials and methods section does not make this sufficiently clear (it practically does not give complete information), the results' interpretation is impaired.
For example:
Item 3.2 and 3.3: How were the Bifidobacterium isolates identified? What does this number represent? Genes or genomic DNA were deposited in some public database? This information is essential here.
Item 3.3: It is unclear how the experiments were done: "Then, the temperature of the shaker was lowered to 18 °C, 200 μL IPTG (1 mol/L) was added to the medium after 30 min of incubation, and the culture was incubated for 48-72 h … the use of the time of 48-72h of expression does not make sense thinking about the expression system used ( pET28a and E. coli BL21). What was the final concentration of IPTG used?
The enzymes used were purified, correct? What is this supposed to mean? In figure 1, it is not clear that the preparations are purified! Why was only one imidazole concentration used and not a gradient in the purification process? In line 70, the authors said: "B17_2 and B26_3 was 79.82 kDa and 78.14 kDa"… How were these values estimated? Based on SDS? was an Rf graph made? How was the expression of the enzymes confirmed? mass spectrometry? Again, it is unclear how the experiments were done!
There are no data reported about the enzyme activity (U mg or U mL) after purification and before purification. Thus, there is no data about the efficiency of the process! another critical point is the concentration of proteins in the solution after purification (line 74: 34.8 mg/mL, 17.8 mg/mL, and 21.92 mg/mL)... weren't the proteins precipitated in solution?
Item 3.10. Sorry, but I needed to re-read the manuscript a few times to understand what was done partially. "Lactose hydrolysis was determined using the 3, 5-dinitrosalicylic acid DNS assay"… Sorry, Do you mean: the reaction is stopped using 3, 5-dinitrosalicylic acid? What is this supposed to mean? Are you talking about the Muller method? What were the reaction conditions and the other test conditions? How were the controls carried out? The information given here is insufficient for the reader to know exactly what was done!
(3) In the results section, it is hard to understand the real significance of what the authors have done and the interpretation of the results.
Thus, it is complicated to understand and validate the data presented in the manuscript.
Reviewer 2 Report
The manuscript describes relevant information regarding the clonation and expression of two β-galactosidases belonging to the Bifidobacterium GH42 family for lactose hydrolysis. The introduction provides all the necessary information, the material and method section can be easily reproduced and the results and discussion section is well presented and sufficiently discussed. Some minor comments: - please put the microorganism names in italics i.e. line 107, 236, 248. Revise the whole manuscript. - Line 344 - please inficate the instrument used for the absorbance measurement - in the material and method section the methods described in subsections 3.6 - 3.9 are based on...? - in the conclusion please indicate some further studies or directions of this study Overall the manuscript is very good and after some minor corrections it can be published.
Reviewer 3 Report
In this manuscript, the authors molecular characterized two beta-galactosidase in Bifidobacterium. The following comments should be addressed.
- Page 1, lines 7-13: E-mail of all authors should be provided.
- Page 1, line 16 and line 19: Mild suggestion, B17_2 and B26_3 could be replaced by BlGLB1 and BpGLB1, respectively. Gene names provide much more info than just number. BlGLB1 stands for Bifidobacterium longum beta-galactosidase 1.
- Page 1, line 21: oNPG is an abbreviation. Try not to use abbreviation in abstract.
- Page 1, line 29: GH42 is an abbreviation and also odd here. Please consider if it should be removed or not?
- Page 1, lines 41-42: The global production of whey ……..200 million tons. First, reference is needed. Second, it was already 14 years ago, can authors provide new info?
- Page 2, line 66: B of Bifidobacterium should be italic.
- Page 2, line 69: The authors should explain what are B17_2 and B26_3?
- Page 2, lines 74-75: the protein concentration of three beta-galactosidase are comparable. However, in Figure 1 legend (line 116), concentration of commercial enzyme E is significantly lower than that of other beta-galactosidase. Both numbers are conflicted but maybe corrected. As we can see in Figure 1a, in the legend, authors described three enzymes (line 114), but only two protein bands were shown in Figure 1a, please explain. To me, it looks like B26_3 is enzyme E and one protein band, maybe B26_3, is missing, considering B17_2 and B26_3 have similar molecular weight. By the way, the original SDS-gel (supplementary result) have three protein samples.
- Fig 1, please place figure and legend on the same page.
- Page 3, line 104 and Page 4, Table 1: The presence of Ni ion significantly increased the activity of B26_3 by 2127%. I think it is a typo, but if not, please explain.
- Page 3, line 107: Penicillium canescens should be italic. Also, Page 4, line 131: Bifidobacterium should be italic
- Page 4, Table 1: Note, about the superscript ‘2’, I cannot find it in the Table 1. Also, Table 2, Note, about the superscript ‘3’, I cannot find it in the Table 2. In addition, Table 3, Note, about the superscript ‘4’, I cannot find it in the Table 3.
- Page 5, Figure 2: Mild suggestion, subfigures a-c, [S] can be replaced by [oNPG] and subfigures d-e, [S] can be replaced by [lactose].
- Page 6, Table 2: About reference, a mild suggestion, use ‘this study’ to replace B17_2 and B26_3.
- Page 6, lines 152-154 and Fig 3b: Is this result (three-fold increased activity) repeatable? I mean, how many individual experiment(s) were performed?
- Page 8, Table 3: Some results are odd. For example, the specific activity of using oPNG by B17_2 enzyme is 130 plus and minus 90. First, why the standard deviation is so big? Second, if you go back to Figure 2a (page 5), the Vmax is 3700 plus and minus 100. Please double check the result.
- Page 8, Table 3: Another conflicts: line 181, the author described the activity of E against lactose was 2.3 plus minus 1.6. First, explain why the standard deviation is so big. Second, in Table 3, the activity of E against lactose was 1.6 plus minus 0.1. The activity result of 2.3 plus minus 1.6 was found in enzyme E against oPNG. Please confirm it.
- Page 9, Figure 4: Mild suggestion, place figure and legend on the same page.